# Electrochemical Determination of Dopamine with a Carbon Paste–Lanthanum (III) Oxide Micro-Composite Electrode: Effect of Cetyl Trimethyl Ammonium Bromide Surfactanton Selectivity

**DOI:** 10.3390/s24165420

**Published:** 2024-08-22

**Authors:** Edgar Nagles, Fernando Riesco, Luz Roldan-Tello

**Affiliations:** Facultad de Química e Ingeniería Química, Universidad Nacional Mayor de San Marcos, Lima 15081, Peru; fernando.riesco@unmsm.edu.pe (F.R.); luz.roldan@unmsm.edu.pe (L.R.-T.)

**Keywords:** carbon paste, dopamine, ascorbic acid, lanthanum oxide, square wave voltammetry

## Abstract

This paper presents a new application of a lanthanum oxide (III)-modified carbon paste electrode (La_OX_/CPE) for dopamine (DP) detection in the presence of ascorbic acid (AA). The presence of cetyl trimethyl ammonium bromide (CTAB) facilitated the La_OX_/CPE electrode’s ability to detect DP amidst AA interference, resulting in a substantial 70.0% increase in the anodic peak current for DP when compared to the unmodified carbon paste electrode (CPE). CTAB enabled clear separation of the anodic peaks for DP and AA by nearly 0.2 V, despite their initially overlapping potential values, through the ion–dipole interaction of AA and CTAB. The electrode was characterized using cyclic voltammetry (CV) and energy-dispersive spectroscopy (EDS). The method demonstrated a detection limit of 0.06 µmol/L with a relative standard deviation (RSD) of 6.0% (*n* = 15). Accuracy was assessed through the relative error and recovery percent, using urine samples spiked with known quantities of DP.

## 1. Introduction

Dopamine, a catecholamine and one of the most prevalent neurotransmitters in humans, plays a critical role in various brain functions including consciousness, stress, attention span, learning, motivation, movement, and memory [1,2]. The degeneration of dopaminergic neurons, often attributed to oxidative stress, significantly impacts neuronal health. In response, the World Health Organization (WHO) initiated a 10-year plan in 2022 to address neurological disorders, including epilepsy and Parkinson’s disease, which has seen an 80% increase in prevalence since 2000 and now affects over 8.5 million individuals [3]. The WHO’s treatment strategy involves the use of Carbidopa/levodopa to elevate brain dopamine levels. An accurate measurement of the dopamine levels in blood is essential before starting this medication, with normal levels ranging from 0.01 to 0.10 µmol/L [4].

Dopamine (DP), uric acid (UA), and ascorbic acid (AA) commonly coexist in biological samples, and their measurement is important due to their implications for human health [5,6,7,8]. High-performance liquid chromatography (HPLC) is typically used for their detection, though it cannot simultaneously measure these compounds. HPLC coupled with electrochemical detection (HPLC-ECD) offers high sensitivity (picomolar range) but is complex and costly. Electroanalytical techniques present a more convenient and cost-effective alternative for simultaneous detection, offering high sensitivity and selectivity while eliminating the need for extensive sample preparation.

Despite advancements, detecting DP, UA, and AA simultaneously remains challenging due to their overlapping oxidation potentials, leading to interference [9]. With the development of new materials, researchers are utilizing them for the individual and, in some cases, simultaneous detection of DP, UA, and AA. Materials such as poly-diaminopyrene [10], poly-vinyl alcohol [11], carbon nanotubes [9,12], graphene [13,14], and metal cation nanoparticles (e.g., gold [15] and platinum [16] nanoparticles) have been used for the simultaneous detection of DP, UA, and AA. However, studies on independent detection are more abundant. In 2016, a review was published detailing the most commonly used materials for the simultaneous detection of DP, UA, and AA, indicating that metal cation nanoparticles are the least used, with only gold, platinum, palladium, and titanium nanoparticles being reported [17]. In 2019, another review was published which reported an increase in the use of nanomaterials [5], but the use of metal cations remained scarce. In 2023, the latest review to date reported on the multiple detection of DP, UA, and AA with molecularly printed polymer (MIP) electrodes [18]. The review found that while MIP electrodes are very selective and sensitive, they are also expensive and complex to produce. In more recent years, the use of metal oxides such as ruthenium oxide and tin oxide has increased for simultaneous detection. In the first reports, ruthenium oxide [19] and tin oxide [20] were used for UA and AA detection, respectively. There are several reports on other metal oxides, such as magnesium and iron, but these are combined with other nanomaterials, such as graphene. Reports on the use of metal oxides supported on carbon for the simultaneous detection of DP, UA, and AA without the use of other materials are still limited.

The use of lanthanum oxide to modify carbon paste electrodes has increased in recent years due to its electrocatalytic properties [21]. In 2019, La_2_O_3_ was employed in conjunction with carbon paste for the first time to detect thimerosal, followed by its application in 2020 for the detection of paracetamol; the surface of the modified electrode was characterized by SEM and EDS in both instances. In 2021 and 2023, La_2_O_3_ combined with TiO_2_ on carbon paste was used to detect tartrazine, sunset yellow, and allura red [22,23]. The present study builds upon prior research by calculating the active surface area, which was not previously reported.

The detection of dopamine using carbon electrodes modified with lanthanum has been previously reported. For this purpose, various forms of lanthanum have been used in combination with different materials, such as LaN_3_O_6_·6H_2_O with carbon nanotubes [24], LaCl_3_ with graphene oxide [25], lanthanum ortho ferrite nanoparticles (LaFeO_3_) [26], a lanthanum metal–organic framework (La-BTC) combined with carbon nanotubes [27], La_2_O_3_ combined with AuNPs–tyrosinase enzyme [28], and AgNPs with graphene oxide [29]. The reports using lanthanum yielded calculated detection limits ranging from 0.60 to 0.013 µmol/L, demonstrating the significant impact of lanthanum on dopamine electro-oxidation. The present study introduces a simpler and more easily manufactured electrode compared to previous complex designs.

The role of surfactants in electrochemistry has been reviewed [30], highlighting their advantages such as their low cost, environmental friendliness, and good solubility. These benefits support the use of cetyl trimethyl ammonium bromide (CTAB) in this study.

## 2. Materials and Methods

### 2.1. Reagents and Instruments

Water for the preparation of all solutions and the electrochemical cell was obtained from a Thermo Sciences water purification system (Madrid, Spain), and phosphate buffer solution (PBS) was prepared from H_3_PO_4_, NaH_2_PO_4_ and Na_2_HPO_4_ salts from Merck (German Burlington, MA, USA). DP, AA, UA, K_3_/K_4_Fe(CN)_6_, KCl, CTAB, SDS, CPB, carbon paste, and La_2_O_3_ were obtained from Sigma-Aldrich (Burlington, MA, USA). Cyclic voltammetry (CV) and square wave voltammetry (SWV) measurements were obtained with a Dropsens µ-Stat 400i potentiostat from Metrohm (Herisau, Swiss), and energy-dispersive spectroscopy (EDS) was performed using a JEOL model JSM 6490-LV (Tokyo, Japan) with a secondary electron detector. A Ag/AgCl electrode and a platinum wire from Chi-instrument (New York, NY, USA) were used as reference and auxiliary electrodes, respectively.

### 2.2. Measurements

The active surface area of the modified and unmodified carbon paste electrodes with La_2_O_3_ was calculated using K_4_Fe(CN)_6_ 5.0 mmol/L, and the electrochemical behavior of DP and AA was evaluated. Optimal parameters were identified, and calibration curves for standard solutions and real samples were developed using SWV with the following optimal parameters: E_Step_, 10.0 mV; E_AMP_, 100.0 mV; E_ACC_, −0.20 V; t_ACC_, 30.0 s; and PB pH, 7.0. The electrochemical cell was completed using 9.0 mL of HPLC-grade water, 0.5 mL of PBS, and 250.0 µL of DP and AA at 10.0 mmol/L (0.25 mmol/L in solution) by CV. For all SWV studies, 10.0 to 100.0 µL of a 1.0 mmol/L DP solution, corresponding to 1.0 to 10.0 µmol/L in the final solution, was used.

### 2.3. Sample Treatment

The human urine used to evaluate the accuracy of DP detection was donated by a laboratory collaborator. The real sample was spiked and did not require prior treatment before analysis. Only 1.0 or 2.0 mL of the sample was used. This amount was necessary to obtain a more complex matrix and evaluate the interference of uric acid and metal cations present in urine.

### 2.4. Preparation of CPE and La-OX/CPE

The detection of DP was studied using two electrodes: CPE and La_OX_/CPE. The unmodified CPE was prepared with 50.0 mg of carbon paste compacted with 10.0 µL of mineral oil in a PVC cylinder with a length of 10.0 cm and an internal diameter of 3.0 mm. The compacted paste had a height of 1.0 mm. A steel wire was used as the contact. The La_OX_/CPE electrode was modified with minor adjustments to previous reports [31], using the same amount of carbon paste and mineral oil. However, before being compacted inside the PVC cylinder, it was mixed with La_2_O_3_ (between 1.0 and 5.0 mg) in an agate mortar. The optimal amount of La_2_O_3_ used in all measurements was 3.0 mg. An increase in the amount of La_2_O_3_ did not result in significant changes in the anodic current values. Both electrodes underwent three cycles of CV between 0.0 and 1.0 V at a scan rate of 0.1 V/s to ensure a homogeneous surface prior to each measurement.

## 3. Results and Discussion

### 3.1. Electrode Characterization and Dopamine Interaction Study

Figure 1 shows the cyclic voltammograms (Figure 1A,B) and square wave voltammograms (Figure 1C,D) without and with DP. It can be clearly observed that DP is oxidized at more positive potential values by CV compared with SWV. The significant increase in the anodic current for DP with the electrode modified with La_2_O_3_ is also apparent (red curves in Figure 1A,B). The anodic peak currents and potential values are summarized in Table 1. The results indicate that the anodic current for DP is higher at lower concentration values when measured by SWV, and the increase in current is greater than 500% when compared to the electrode without La_2_O_3_. Previous studies have observed the oxidation of DP at potential values very close to the new reported value (between 0.2 and 0.3 V) with modified electrodes such as Fe_3_O_4_ [32], cobalt–tannic acid [33], and PtAu hybrid film [34]. These results suggest that the lanthanum oxide composite exhibits similar properties to the other nanostructured oxides used for the oxidation of DP.

These results demonstrate a clear positive effect on DP oxidation with the La_2_O_3_-modified electrode. To investigate the cause of this effect, we evaluated the electrode’s active surface, which has not been previously calculated or reported.

In the case of the electrodes without DP (Figure 1A,B), it is clear that the presence of La_2_O_3_ does not exhibit electroactivity between 0.0 and 1.0 V, creating a clean baseline that does not interfere with the DP signal. However, an increase in the background current is evident for the modified electrode, which is likely attributable to the increased active surface area of the electrode.

The electrode area was calculated using the [Fe(CN)_6_]^3+/4+^ (5.0 mm/L) redox system with KCl (1.0 mol/L). Figure 2A,B show the voltammograms for CPE and La_OX_/CPE, respectively, with scan rates ranging between 0.02 and 0.16 V/s. In addition, Figure 2C illustrates the plot of the anodic peak current (Ip(A)) versus the square root of the scan rate (ʋ^1/2^) for both CPE (black line) and La_OX_/CPE (red line). These results reveal a nearly twofold increase in the anodic peak current across all scan rates with La_OX_/CPE, while the anodic peak potentials remain relatively constant, resulting in a consistent ΔV for both electrodes. These trends are consistent with those observed for DP.

The active surface area of each electrode was calculated using the Randles–Sevcik equation.
I*p* = 2.69 × 10^5^ n ^3/2^ AD^1/2^ ν ^1/2^C

The calculated values, based on the value of the slope of the Ip/square root of the scan rate (Ip(A)/ ʋ ^1/2^V/s) of the anodic and cathodic peak currents shown in Figure 2C and taking into account the concentration (5.0 mmol/L) and diffusion coefficient D (7.6×10^−6^ cm^2^s^−1^) for the [Fe(CN)_6_]^3+/4+^ redox system, are summarized in Table 2. The electrode modified with La_2_O_3_ displays a significant increase in the active surface area compared to the unmodified electrode. The effect observed in Figure 1A and Figure 2B may be attributed to the increase in surface area and electrostatic interactions. The calculated active surface area value was found to be higher compared to those of other lower-molecular-weight oxides, such as molybdenum oxide [35], but lower compared to those of electrodes modified with organic compounds like chitosan [36].

SEM results for La_OX_/CPE are shown in Figure 3A. Small white crystals of varying sizes are clearly visible and evenly distributed over the carbon surface. One of the crystals (spectrum 1) was measured by EDS to calculate its composition, which is shown in Figure 3C. The EDS analysis of the CPE (Figure 3B) shows the presence of C without oxygen, and the EDS image of the La_OX_/CPE microstructure (Figure 3A) shows the presence of C, La, and O. Furthermore, the percentage weight ratio of O/La given by the EDS analysis (Figure 3C) is 0.18. The theoretical value is 0.17 for La_2_O_3_. This confirms the presence of only La (III) on the surface of the modified electrode.

### 3.2. pH Effect

Figure 4 shows the electroactivity of DP as a function of pH using La_OX_/CPE. According to the results, the maximum anodic peak current for DP with La_OX_/CPE was observed at pH 7.0 using PB. This result suggests that at pH 7.0, DP is more easily deprotonated due to its high pKa value. The anodic peak potential shifted to less positive values with increasing pH, ranging from 0.59 V at pH 2.0 to 0.30 V at pH 7.0, as shown in Figure 4A,B, presenting a linear relationship for the plot of Ep(V) vs. pH with Ip(µA) = 0.7308 − 0.056pH (R^2^ = 0.989). The slope equation is m = 0.059H^+^/n, where n is the number of electrons for a reaction, and H^+^ = e^−^. The slope has a value of 0.056 for a reaction where 2H^+^ = 2e^−^. Typically, DP exhibits high values of anodic peak currents at acidic pH values ranging from 2.0 to 4.0. This is contrary to previous reports where Nd_2_O_3_ with a molecular weight similar to that of La_2_O_3_ and glassy carbon electrodes (GCEs) were used and the maximum anodic currents were observed at pH 3.0 [37,38]. Additionally, electrodes modified with substances such as carbon nanotubes [38] and metal cation complexes [39] showed the highest anodic current at an acidic pH of 2.5.

### 3.3. Scan Rate Effect on DP Using La_OX_/CPE

Identifying the mass transfer of DP at the electrode interface is critical to evaluating the lifetime of the modified electrode. Diffusion-controlled processes allow for multiple uses of the electrode, whereas adsorption-controlled processes lead to the development of surface memory effects caused by the adsorbed analyte. If the process is controlled by adsorption, it is necessary to perform treatments on the electrode surface to desorb the adsorbed analyte prior to reuse. The study of mass transport for DP on La_OX_/CPE was evaluated by CV, with the scan rate varying between 0.02 and 0.20 V/s. Figure 5 shows the voltammograms and plots of Ip (µA) vs. the square root of the scan rate (V/s) and log Ip(µA) vs. log scan rate (V/s). The results show that the anodic peak currents increase proportionally with slow scan rates, between 0.02 and 0.08 V/s. At higher scan rates, between 0.10 and 0.2 V/s, the increase in anodic peak currents is smaller (Figure 5A). The plot of Ip (µA) vs. the square root of the scan rate (V/s) shows an R^2^ of 0.995 (Figure 5B). This linear relationship confirms that mass transfer at the electrode surface is a diffusion-controlled process. Additionally, the plot of log Ip (µA) versus log scan rate (V/s) (Figure 5C) shows a slope of 0.475 (R^2^ = 0.992), which is close to the theoretical value of 0.50 for a diffusion-controlled process. This finding may explain why the current for DP in Figure 6F does not increase in each cycle, since no adsorption processes are taking place.

### 3.4. Optimization of DP Parameters and Assessment of the Lifespan of La_OX_/CPE

With the aim of increasing the DP anodic peak current using La_OX_/CPE and achieving a low detection limit and good sensitivity, other parameters were investigated. We evaluated the accumulation potential (E_ACC_) and accumulation time (t_ACC_) during the accumulation stage, as well as the amplitude potential in mV (E_AMP_) and step potential (E_Step_) during the stripping stage by SWV. The measurements were conducted using the univariate method, where only one parameter was varied at a time. Parameters such as the frequency (Hz) were not considered, as no significant variation in the DP anodic peak currents was observed between 10 and 50 Hz. The plots of Ip (µA) versus E_Step_ (Figure 6A), E_AMP_ (Figure 6B), E_ACC_ (Figure 6C), and t_ACC_ (Figure 6D) demonstrate a proportional increase in the anodic peak current for DP at 0.04 mmol/L. The current increased from 6.5 to 10.0 µA with an E_Step_ of 10.0 mV, from 10.0 to 20.0 µA with an E_AMP_ of 100.0 mV, from 20.0 to 22.5 µA with an E_ACC_ of −0.20 V, and from 22.0 to 25.0 µA with a t_ACC_ of 30.0 s. Figure 6E shows the DP anodic peak currents with non-optimized parameters (black curve) and optimized parameters: EStep of 10.0 mV, E_APM_ of 100.0 mV, E_ACC_ of −0.20 V, t_ACC_ of 30.0 s, and pH of 7.0 PB. It is clear that the current increased by almost 100% without a change in the DP oxidation potential value. This improvement allows for a lower detection limit to be achieved.

To evaluate the stability of La_OX_/CPE, 50 consecutive CV cycles with DP at 0.25 mmol/L and a scan rate of 0.10 V/s were performed (Figure 6F). The voltammograms indicate that there was no increase in anodic peak currents with each cycle, confirming the absence of any adsorption processes. However, the anodic peak currents of the first and last cycles were 25.0 µA and 23.5 µA, respectively, resulting in a 6% difference. This suggests a loss of approximately 0.12% of anodic current per cycle. The average anodic peak current value of the 50 measurements was 24.6 µA with a standard deviation (σ) of 0.510. The coefficient of variation percentage (% CV) was 2.0%. This result implies that the modified electrode retains its activity and is suitable for a range of measurements.

### 3.5. Calibration Curve, Detection Limits, and Reproducibility

The calibration curves were developed in two DP concentration ranges, 0.20–0.95 µmol/L and 0.90–4.5 µmol/L, using La_OX_/CPE with optimal parameters: E_Step_, 10.0 mV; E_AMP_, 100.0 mV; E_ACC_, −0.20 V; t_ACC_, 30.0 s; and PB pH, 7.0 by SWV. The voltammograms are depicted in Figure 7. The calibration curve for the first range revealed a linear relationship of Ip(µA) = −0.54 + 6.11C_DP_ with a detection limit of 0.12 µmol/L (Figure 7A) and a sensitivity of 6.11 µA/µM. In the second range, as shown in Figure 7B, a linear relationship of Ip(µA) = −0.35 + 3.61*C*_DP_ with a detection limit of 0.27 µmol/L and a sensitivity of 3.61 µA/µM was observed. The results indicate that it is possible to achieve detection limits below 1.0 µmol/L and a signal (µA)/concentration (µmol/L) ratio exceeding 1.0 µA/µmol/L in both concentration ranges. Furthermore, the obtained results are comparable to those in previous reports using electrodes modified with metal oxides of different molecular weights than La_2_O_3_ and combined with other substances. Some of these reports are summarized in Table 3. The detection limits reported with these electrodes are almost in the same concentration range as those calculated using this new method.

Reproducibility was studied with three different electrodes using DP at 3.0 µmol/L. The average anodic peak current was 7.90 µA with a standard deviation (σ) of 0.05 and a coefficient of variation percentage (%CV) of 0.60%. This result was much lower than that observed in Figure 6F, possibly due to the fact that the measurements were made at higher concentrations with CV, where the parameters were not optimized compared to those with SWV.

### 3.6. Interference Study with DP, AA, and CTAB

Out of all the substances evaluated for potential interference with the DP signal over La_OX_/CPE, including glucose, uric acid, paracetamol, caffeine, and metallic cations such as Pb, Cd, Na, Fe, Ni, Zn, and Cu, only ascorbic acid (AA) significantly interfered at the same potential value observed for the oxidation of DP. Figure 8 shows the voltammograms for AA and DP, both individually and simultaneously, in the presence of CTAB. The voltammograms revealed that both AA and DP undergo oxidation at the same potential value when La_OX_/CPE is used at pH 7.0 (Figure 8A). Although DP exhibited higher activity, the contribution of AA cannot be disregarded. Furthermore, at a more acidic pH (pH 2.0), both activities were similar, with DP showing slightly higher activity and a separation of almost 0.25 V (Figure 8B). Although it may be assumed that AA’s interference was eliminated at pH 2.0, the individually measured voltammograms revealed overlapping curves. The simultaneous measurement of DP-AA voltammograms showed synergy, causing both signals to overlap at the same potential value (see Figure 8C). Therefore, at pH 2.0, AA interference remains unresolved. To separate the oxidation signals of DP and AA and ensure that they appear at different potential values, the effect of a cationic surfactant, CTAB, was studied. Amphiphiles such as CTAB have been reported to interact with analytes at electrode interfaces through charge and polarity, causing potential peaks to shift to lower energy values. This interaction is due to the chemical structure of amphiphiles, which comprises hydrophobic and hydrophilic zones. The combined use of lanthanum oxide and CTAB for PD detection was previously reported using a GCE modified with carboxylated graphene. With this modified electrode, a detection limit of 0.036 µmol/L was reported [40]. This detection limit is very close to that in this new report, where a simpler electrode was used.

Another anionic surfactant, sodium dodecyl sulfate (SDS), and another cationic surfactant, cetyl peridinium bromide (CPB), did not present the same effect on DP and AA as CTAB. Therefore, this effect depends on the charge and cation of the surfactant.

In this case, CTAB showed a strong affinity for AA over DP. Figure 8D illustrates the voltammogram of AA (black curve) alone and in the presence of CTAB (red curve). The anodic peak of AA shifted to less positive potential values, from 0.65 V to 0.41 V, upon the addition of CTAB. Moreover, the anodic current increased by almost 40%, indicating a significant effect on both the potential and current of AA. Upon the introduction of the same concentration of DP into the cell, the anodic peak of DP appeared at 0.65 V, and the AA and DP signals (blue curve) were clearly distinguishable, as depicted in Figure 8C. Consequently, the AA-DP signals were separated by more than 0.20 V. This effect was also observed through SWV at pH 2.0 and at lower concentrations (Figure 8E). Previous studies have shown that the interaction between CTAB and AA enhances the antioxidant activity of AA, as revealed by tensiometric and spectroscopic methods [41]. In this scenario, the ion–dipole interaction between AA and CTAB enhances the electroactive properties of AA over La_OX_/CPE, thereby eliminating interference in the detection of DP. It is worth noting that the use of surfactants in detecting DP and AA is not a new concept. In 2013, Shahrokhian reported the use of the cationic surfactant Tetra-n-octylammonium bromide (TOAB) [42]. In 1999, Wen reported the utilization of CTAB and sodium dodecyl sulfate (SDS) on GCEs [43]. These surfactants, TOAB, CTAB, and SDS, were employed to improve the resolution of DP and AA signals, thereby enhancing selectivity.

### 3.7. Analytical Application in Real Samples

To assess the versatility of La_OX_/CPE in detecting dopamine (DP), human urine samples were utilized. Human urine, a complex biological matrix, contains various compounds including urea (U), uric acid (UA), ammonia (A), creatinine (CT), and mineral salts. The urine samples were spiked with DP concentrations of 2.0 and 5.0 µmol/L and analyzed using the standard addition method. The analysis revealed DP concentrations of 2.20 ± 0.01 µmol/L (sample 1) and 4.89 ± 0.02 µmol/L (sample 2), with relative errors (% RE) of 10.0% and −2.75%, respectively. Notably, the urine samples underwent no prior treatment. Interestingly, despite the high concentration of uric acid in the urine samples, the UA signal did not interfere with the DP signal, which exhibited a lower potential compared to the UA signal. Moreover, the anodic peak potential of DP in the sample mirrored the values observed in Figure 7, suggesting that the matrix had no discernible impact on La_OX_/CPE activity. Figure 9 displays the voltammograms of the 2.0 µmol/L DP sample alongside the calibration plot (insert). These findings underscore the robustness of La_OX_/CPE in detecting DP within complex biological matrices such as human urine.

## 4. Conclusions

This study presents a novel application of a lanthanum oxide (III) (La_OX_)-modified carbon paste electrode for DP detection in the presence of AA with the aid of CTAB. The modified electrode demonstrated a significant 70.0% increase in the anodic peak current for DP when compared to the unmodified CPE. Despite DP and AA exhibiting anodic peak currents at the same potential, the inclusion of CTAB enabled clear separation by nearly 0.2 V. The La_OX_-modified CPE offers a promising platform for sensitive and selective DP detection in complex sample matrices. The achieved detection limit for DP was 0.06 µmol/L, with an RSD of 6.0% (*n* = 15). The method’s accuracy was assessed using urine samples spiked with known quantities of DP. In summary, the developed La_OX_-modified CPE holds potential as an effective tool for DP detection in practical applications, offering improved accuracy, stability, and reproducibility.

## Figures and Tables

**Figure 1 sensors-24-05420-f001:**
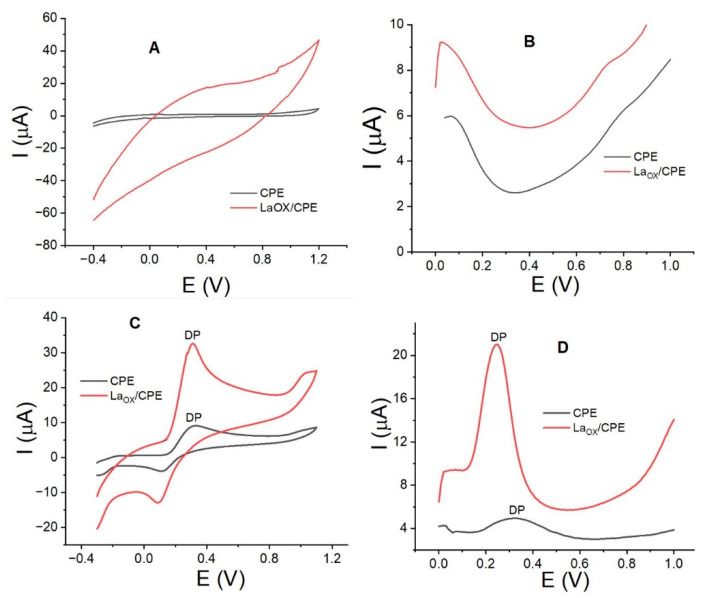
Cyclic voltammograms of CPE (black curve) and La_OX_/CPE (red curve) without DP (**A**) and with DP (0.25 mmol/L) (**C**) at a scan rate of 0.1 V/s. Square wave voltammograms of CPE (black curve) and La_OX_/CPE (red curve) without DP (**B**) and with DP (0.04 mmol/L) (**D**). Experimental conditions: E_Step_, 1.0 mV; E_APM_, 50.0 mV; E_ACC_, 0.0 V; t_ACC_, 60.0 s; PB pH, 3.0.

**Figure 2 sensors-24-05420-f002:**
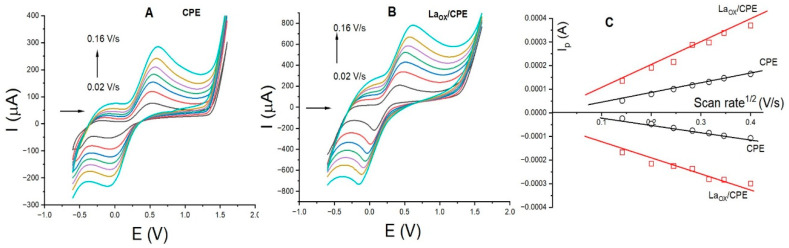
Cyclic voltammograms of [Fe(CN)_6_]^3+/4+^ (5.0 mmol/L) at scan rates between 0.02 and 0.16 V/s with (**A**) CPE and (**B**) La_OX/_CPE; (**C**) plot of the square root of the scan rate vs. the anodic peak current. Experimental conditions: pH 3.0 (PB).

**Figure 3 sensors-24-05420-f003:**
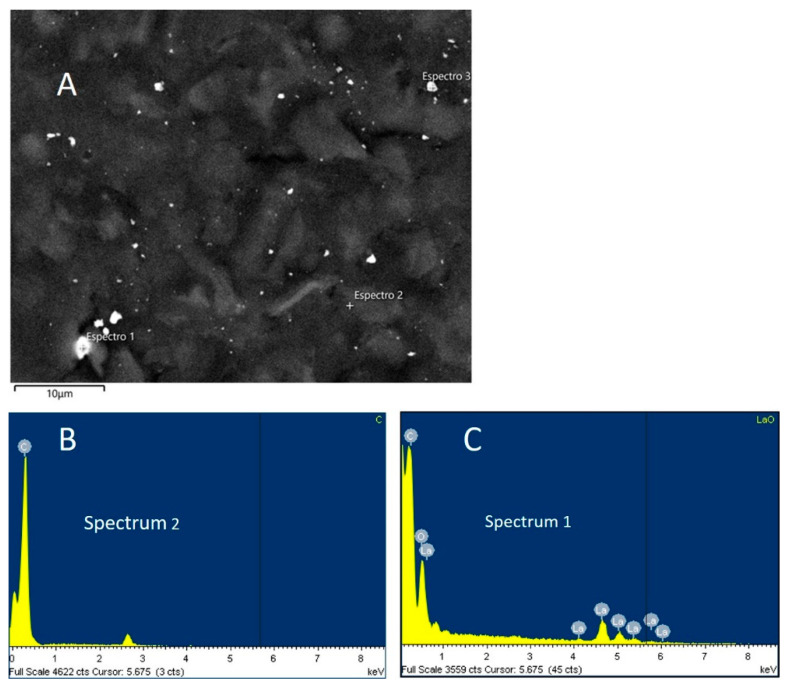
SEM image of La_OX_/CPE (**A**) and EDS spectra of CPE (**B**) and La_OX_/CPE (**C**).

**Figure 4 sensors-24-05420-f004:**
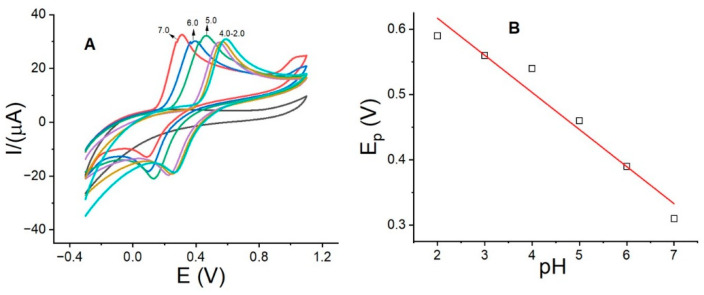
(**A**) Cyclic voltammograms of DP using La_OX_/CPE in PB with pH between 2.0 and 7.0 and (**B**) plot of Ep(V) vs. pH. Experimental conditions: DP, 0.25 mmol/L; scan rate, 0.1 V/s.

**Figure 5 sensors-24-05420-f005:**
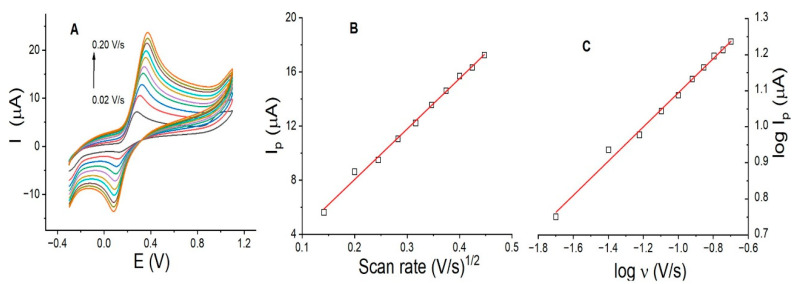
(**A**) Cyclic voltammograms of DP using La_OX_/CPE, (**B**) plot of Ip (µA) vs. ʋ^1/2^ (V/s), and (**C**) plot of log Ip (µA) vs. log ʋ (V/s). Experimental conditions: DP, 0.25 mmol/L; scan rate, 0.02–0.20 V/s; pH 7.0 in PB.

**Figure 6 sensors-24-05420-f006:**
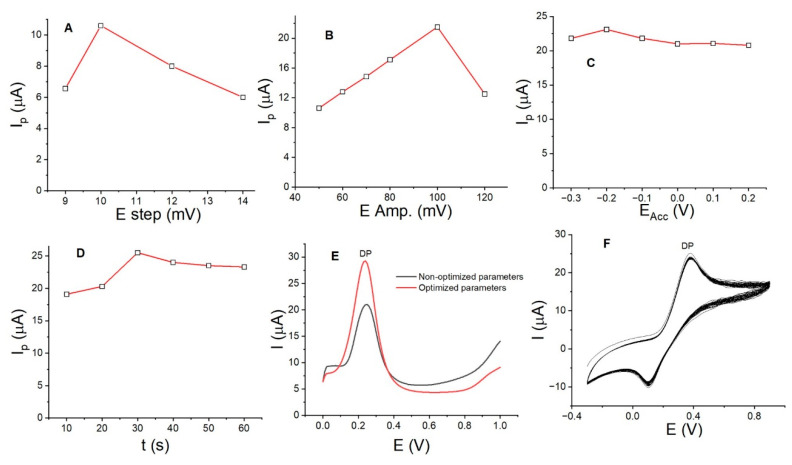
(**A**) Effect of E_step_, (**B**) effect of E_AMP_, (**C**) effect of E_ACC_, and (**D**) effect of t_ACC_ on the anodic peak currents for DP at 0.04 mmol/L. (**E**) SWV for DP with non-optimized parameters (black curve) and optimized parameters (red curve) and (**F**) CV (50 cycles) for DP at 0.25 mmol/L using La_OX_/CPE in PB solution, pH 7.0, with a rate of 100.0 mV/s.

**Figure 7 sensors-24-05420-f007:**
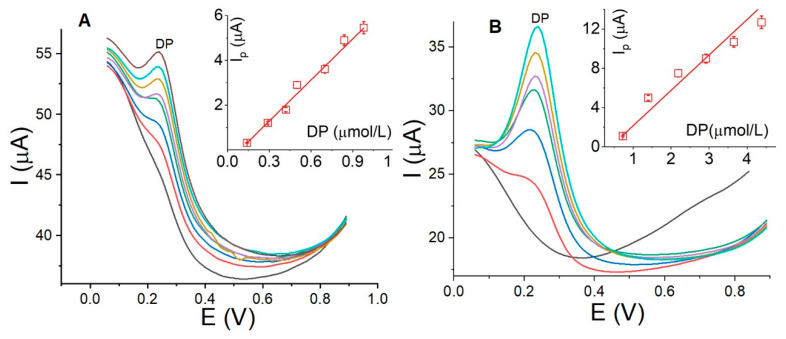
Square wave voltammograms and corresponding calibration curves (insert) depicting DP concentrations ranging from 0.14 to 0.98 µmol/L (**A**) and from 0.73 to 4.3 µmol/L (**B**) using La_OX_/CPE. Experimental conditions: pH 7.0; E_step_, 10.0 mV; E_amp_, 100.0 mV; E_Acc_, −0.2 V; t_Acc_, 30.0 s; frequency, 40.0 Hz.

**Figure 8 sensors-24-05420-f008:**
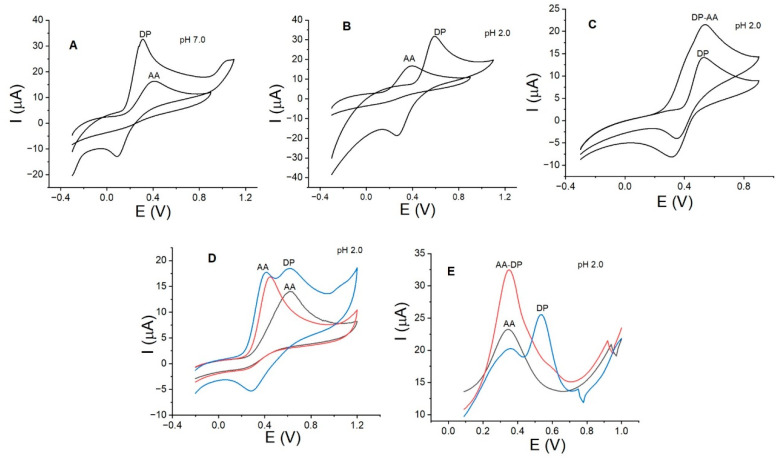
Cyclic voltammograms illustrating DP and AA concentrations of 0.25 µmol/L at (**A**) pH 7.0 and (**B**) pH 2.0, and (**C**) cyclic voltammograms depicting simultaneous DP-AA detection at pH 2.0. The impact of CTAB on the anodic peak currents for AA and DP is demonstrated by (**D**) cyclic voltammetry and (**E**) square wave voltammetry, with AA shown in black curves, AA with CTBA in red curves, and DP-AA with CTAB in blue curves. The experimental conditions are similar to those in Figure 7.

**Figure 9 sensors-24-05420-f009:**
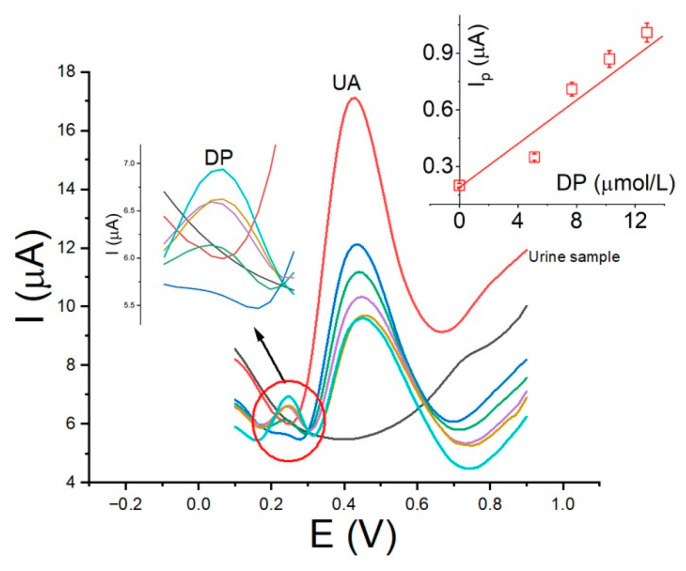
Square wave voltammograms for urine sample 1 using La_Ox_/CPE, with the calibration plot inserted. The conditions are consistent with those described in Figure 7.

**Table 1 sensors-24-05420-t001:** The anodic peak current increases for DP with CPE and La_OX_/CPE.

Electrode	CV		% Current Increased	SWV		% Current Increased
	E_p_ (V)	I_p_/(μA)		E_p_ (V)	I_p_/(μA)	
CPE	0.31	5.0		0.32	1.03	-
La_OX_/CPE	0.32	22	400	0.24	11.2	900

**Table 2 sensors-24-05420-t002:** Active surface areas of CPE and La_OX_/CPE.

Electrode	Slope (I_p_/ ʋ ^½^)		Area (cm^2^)		Total Area (cm^2^)
	Ipa	Ipc	Ipa	Ipc	
CPE	0.0004	0.0003	0.054	0.040	0.094
La_OX_/CPE	0.0009	0.0005	0.121	0.067	0.188

**Table 3 sensors-24-05420-t003:** Modified electrodes for DP.

Electrode	Application	LoD (µmol/L)	Ref.
LaMnO_3_ NPs/GCE	Drugs	0.032	[1]
Au NPs Pt NPs	Urine	0.075 0.062	[15]
Ru_Ox_·nH_2_O/GCE		0.080	[19]
GNP/FTO	Human serum	0.220	[22]
La–MWCNTs/CGE GO-COOLa/GCE LaFeO_3_/CPE La-BTC-CNT Tyr/Au-La_2_O_3_/ITO	Drugs Human serum and urine	0.013 0.058 0.600 0.073 0.258	[24] [25] [26] [27] [28]
CTAB/GO-COOLa/GCE		0.036	[40]
La_OX_/CPE	Urine	0.030	This work

GCE: glassy carbon electrode; CPE: carbon paste electrode; GO: graphene; GNP/FTO: graphene nanoplatelet-modified fluorine-doped tin oxide electrode; GO-COOLa: carboxylated graphene oxide/lanthanum; La-BTC-CNT: lanthanum metal–organic framework and carbon nanotube; Try; tyrosinase; ITO: indium tin oxide.

## Data Availability

All data will be provided upon reasonable request to the corresponding author.

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
