# Peer review of "Electrochemical Determination of Dopamine with a Carbon Paste–Lanthanum (III) Oxide Micro-Composite Electrode: Effect of Cetyl Trimethyl Ammonium Bromide Surfactanton Selectivity"

_sensors, 2024, doi:10.3390/s24165420_

Round 1

Reviewer 1 Report

Comments and Suggestions for Authors

In this manuscript, Edgar Nagles and co-workers have reported ‘Electrochemical determination of dopamine with a carbon paste-lanthanum (III) oxide micro-composite electrode. Effect of surfactant on selectivity’. The manuscript can be accepted after carefully revising it with major comments.

1.     The title contains one phrase and one sentence. The authors need to modify it as ‘Electrochemical determination of dopamine with a carbon paste-lanthanum (III) oxide micro-composite electrode: Effect of surfactant on selectivity’.

2.     There is a redundant sentence in the abstract part. Please check it carefully.

3.     The grammatical arrangements of the Experimental part need to be corrected

4.     Some of the chemical formulas of compounds are not correct like LaN3O6·6H2O, La2O3, and others.

5.     The authors need to fully characterize the material by SEM, TEM, XRD, and XPS.

6.     CV and SWV in ‘Electrode Characterization and Dopamine Interaction Study’ were studied only for CPE and LaOx/CPE with DP. The authors need to show only LaOx with and without DP, CPE, and LaOx/CPE without DP.

7.     Why is the potential becoming less positive upon increasing pH from Figure 4A?

8.     The authors have used cationic CTAB surfactant for their study. Why you have only chosen cationic surfactant? Have you tested anionic surfactant like SDS? They need to compare the results of at least CTAB and SDS.

9.     Why do you prefer the electrochemical to the colorimetric method for dopamine determination? Could you prove it?

10.  The authors need to show the reaction mechanisms for the oxidation of DP with possible reaction intermediates from theoretical calculations.

Comments on the Quality of English Language

ut supra

Author Response

The authors are grateful for the revision and recommendations to improve the quality of the manuscript titled “Electrochemical determination of dopamine with a carbon paste-lanthanum (III) oxide micro-composite electrode. Effect of surfactant on selectivity”

The changes in the manuscript were highlighted with yellow color.

Review 1

In this manuscript, Edgar Nagles and co-workers have reported ‘Electrochemical determination of dopamine with a carbon paste-lanthanum (III) oxide micro-composite electrode. Effect of surfactant on selectivity’.The manuscript can be accepted after carefully revising it with major comments.

  1. The title contains one phrase and one sentence. The authors need to modify it as ‘Electrochemical determination of dopamine with a carbon paste-lanthanum (III) oxide micro-composite electrode:Effect of surfactant on selectivity’.

Reply: A colon was added to the title to avoid having two disjointed sentences. However, we retained the name of the surfactant, as the reviewer's proposed title is essentially the same as the original.

  1. There is a redundant sentence in the abstract part. Please check it carefully.

Reply: The abstract has been modified and reduced

  1. The grammatical arrangements of the Experimental part need to be corrected

Reply: The grammatical arrangements of the Experimental section have been reviewed and corrected.

  1. Some of the chemical formulas of compounds are not correct like LaN3O66H2O, La2O3, and others.

Reply: Errors have been corrected throughout the text

  1. The authors need to fully characterize the material by SEM, TEM, XRD, and XPS.

Reply: We attempted to characterize the surface with available instruments at the university. Unfortunately, we do not have all the required techniques, and external studies were inconclusive due to poor execution. However, we believe SEM and EDS characterization is sufficient as the true electroanalytical application cannot be verified solely by surface topography. We have added the SEM image for the electrode modified with La2O3 since the carbon image does not provide any new insights and is well-known.

  1. CV and SWV in ‘Electrode Characterization and Dopamine Interaction Study’ were studied only for CPE and LaOx/CPE with DP. The authors need to show only LaOx with and without DP, CPE, and LaOx/CPE without DP.

Reply: This is a great suggestion. Figures without DP for CV and SWV were added as fig. 1A and fig. 1B. However, preparing an electrode with LaOx alone is not feasible as the oxide powder requires a substrate. By itself, it is not an ideal material. LaOx improves the electroactive activity of carbon. The text discussion has been updated with these new figures.

  1. Why is the potential becoming less positive upon increasing pH from Figure 4A?

Reply: This effect is characteristic of organic molecules that behave as weak acids. To the best of our knowledge, the majority of reports (using a wide variety of electrodes) show that the pH effect has the same tendency.

  1. The authors have used cationic CTAB surfactant for their study. Why you have only chosen cationic surfactant? Have you tested anionic surfactant like SDS? They need to compare the results of at least CTAB and SDS.

Reply: The study compared SDS and another cationic surfactant such as CPB (cetyl pyridinium bromide). These surfactants did not present the activity observed with CTAB. We did not initially mention it but it have been added to the text in this revision.

  1. Why do you prefer the electrochemical to the colorimetric method for dopamine determination? Could you prove it?

Reply: We prefer the electrochemical method due to its sensitivity. For instance, the amount of dopamine in the urine sample was undetectable by spectroscopy, while our electroanalytical approach detected it. Additionally, our line of research focuses on electroanalytical methods.

  1. The authors need to show the reaction mechanisms for the oxidation of DP with possible reaction intermediates from theoretical calculations.

Reply: The redox mechanism is consistent with previous reports and was not initially shown for this reason. The system gave a value close to the theoretical Nernst value for a system where the number of electrons equals the number of protons. If this value had differed, we would be dealing with a different redox system, necessitating the inclusion of the possible redox reaction mechanism.

Reviewer 2 Report

Comments and Suggestions for Authors

The article “Electrochemical determination of dopamine with a carbon paste-lanthanum (III) oxide micro-composite electrode. Effect of surfactant on selectivity” by Nagles et al. is a well-written article on the development of POC devices for healthcare applications. The article can be published in the journal Sensors after addressing the following points:

1. The authors should consider adding a scheme/graphical abstract showing the working principle and highlighting the novelty of this work.

2. Only one cationic surfactant, CTAB, is used in the study. So, “Effect of surfactant on selectivity” should be removed from the title. The current title gives the illusion of a comparative study of different surfactants on the selectivity of the electrochemical study. The authors should mention the mechanisms of surfactant action in the abstract for enhanced selectivity (adsorption, micelle Formation, electrostatic interaction, etc.).

3. Edit what is written on the lefthand side of 1st page above the copyright logo.

4. In the introduction section, add a reference to this sentence “In 2019, another review was published… but the use of metal cations remained scarce”. Also, the authors should mention that HPLC is an extremely sensitive instrument, and it is routinely used for UA, DA, and AA detection in combination with electrochemical sensing (HPLC-ECD) for picomolar sensing. The authors have mentioned the use of metal nanoparticles and La oxides in biosensing applications, but there is a clear gap on why La or La oxides are superior or of interest w.r.t biosensing. What is the novelty of this study in comparison to articles mentioned in reference no. 21, 22, and 23, and what is the difference from Ref (24-29)? Also, mention if there are any reports of La oxide with CTAB for biosensing applications. Add in the last paragraph of the introduction, why this study is important, key points in materials and methods, and key observations/results for summarizing this section.

5. Merge Figures 1 and 2 for better clarity. Figures 4, 5, and 6 show good results on the characterization of the electrode for DA sensing. Were the electrodes washed with DI water after each cycle of measurement in Fig 6(F)? If not, will washing it increases the variation in the outcome?

6. Figure 8 is not ideal for the development of a dopamine sensor. Mention how it can be resolved. There are many examples of separation of the peak, even in the presence of other metabolites (a. https://doi.org/10.1021/acsomega.7b00681; b. https://doi.org/10.1016/j.jallcom.2020.155873; c.  https://doi.org/10.1016/j.bios.2012.10.080).  

7. In the legend for Figure 8, remove the full form of CTAB, make sure to use the full form of any abbreviation only once throughout the manuscript, and make corresponding edits where necessary.

8. Are the electrodes tested directly in undiluted urine? What are the storage criteria for urine? It is a good result to see the DA peak distinct from UA peak in urine samples. Is the tested range of dopamine in urine clinically relevant?

9. Authors may cite these articles in the introduction for a better understanding of non-enzymatic electrochemical sensing (a. doi.org/10.1016/j.trac.2021.116418; b. doi.org/10.1016/j.microc.2020.105599; c. doi.org/10.1007/s00604-022-05612-y).  

Comments on the Quality of English Language

This is a well-written article with clarity of thoughts. 

Author Response

Ms. Sara Strunjas
Assistant Editor

Sensors

The authors are grateful for the revision and recommendations to improve the quality of the manuscript titled “Electrochemical determination of dopamine with a carbon paste-lanthanum (III) oxide micro-composite electrode. Effect of surfactant on selectivity”

The changes in the manuscript were highlighted with yellow color.

Review 2

The article “Electrochemical determination of dopamine with a carbon paste-lanthanum (III) oxide micro-composite electrode. Effect of surfactant on selectivity” by Nagles et al. is a well-written article on the development of POC devices for healthcare applications. The article can be published in the journal Sensors after addressing the following points:

  1. The authors should consider adding a scheme/graphical abstract showing the working principle and highlighting the novelty of this work.

Reply: A scheme has been prepared and uploaded as a graphical abstract

  1. Only one cationic surfactant, CTAB, is used in the study. So, “Effect of surfactant on selectivity” should be removed from the title. The current title gives the illusion of a comparative study of different surfactants on the selectivity of the electrochemical study. The authors should mention the mechanisms of surfactant action in the abstract for enhanced selectivity (adsorption, micelle Formation, electrostatic interaction, etc.).

Reply: The title has been updated and the effect of CTAB has been included in the abstract.

  1. Edit what is written on the lefthand side of 1stpage above the copyright logo.

Reply: Great observation. The text has been edited.

  1. In the introduction section, add a reference to this sentence “In 2019, another review was published… but the use of metal cations remained scarce”. Also, the authors should mention that HPLC is an extremely sensitive instrument, and it is routinely used for UA, DA, and AA detection in combination with electrochemical sensing (HPLC-ECD) for picomolar sensing. The authors have mentioned the use of metal nanoparticles and La oxides in biosensing applications, but there is a clear gap on why La or La oxides are superior or of interest w.r.t biosensing. What is the novelty of this study in comparison to articles mentioned in reference no. 21, 22, and 23, and what is the difference from Ref (24-29)? Also, mention if there are any reports of La oxide with CTAB for biosensing applications. Add in the last paragraph of the introduction, why this study is important, key points in materials and methods, and key observations/results for summarizing this section.

Reply: A reference has been added as [5]. We have included the statement “HPLC is an extremely sensitive instrument...” in the text. This study represents a new application compared to references 21, 22, and 23, as those studies focus on different substances. Additionally, this study calculates the electrode area, which had not been previously reported. The electrodes in references 24-29 are more complex, while the electrode in this study is simpler to manufacture. A report of lanthanum oxide with CTAB for dopamine was previously published, but it features a different electrode than the one reported in this study. This report has been cited and compared in section 3.5. We had not seen it before. Thank you for the suggestion. https://www.worldscientific.com/doi/epdf/10.1142/S0218625X17500974

  1. Merge Figures 1 and 2 for better clarity. Figures 4, 5, and 6 show good results on the characterization of the electrode for DA sensing. Were the electrodes washed with DI water after each cycle of measurement in Fig 6(F)? If not, will washing it increases the variation in the outcome?

Reply: Figures 1 and 2 are not merged because they represent different solutions with distinct results. The differential pulse (DP) does not affect the results shown in Figure 2, and combining these figures might complicate the explanation of the two separate studies. Regarding Figure 6(F), the electrodes were not washed after each measurement to observe if oxidation products affect electrode stability or saturate the surface. Washing the electrode did not change the results in our tests

  1. Figure 8 is not ideal for the development of a dopamine sensor. Mention how it can be resolved. There are many examples of separation of the peak, even in the presence of other metabolites (a. https://doi.org/10.1021/acsomega.7b00681; b. https://doi.org/10.1016/j.jallcom.2020.155873; c.  https://doi.org/10.1016/j.bios.2012.10.080).

Reply: While the cyclic voltammetry (CV) data show limited peak separation, the square wave voltammetry (SWV) results demonstrate considerable separation, with a significant reduction in the anodic peak of ascorbic acid (AA) compared to dopamine (DP), indicating that the sensor performs well for detecting DP. Despite extensive studies, we could not achieve further separation. However, at AA concentrations below 10.0 µmol/L, no signal for AA is observed, making the sensor effective for low-concentration DP detection.

  1. In the legend for Figure 8, remove the full form of CTAB, make sure to use the full form of any abbreviation only once throughout the manuscript, and make corresponding edits where necessary

Reply: The full form of CTAB has been removed from the legend, and all abbreviations are now defined only once throughout the manuscript as required.

  1. Are the electrodes tested directly in undiluted urine? What are the storage criteria for urine? It is a good result to see the DA peak distinct from UA peak in urine samples. Is the tested range of dopamine in urine clinically relevant?

Reply: We used 1 mL of urine diluted in 8 mL of water for testing, as detailed in section 2.3. The dopamine concentrations in urine are relevant and typically low, as observed in clinical contexts.

  1. Authors may cite these articles in the introduction for a better understanding of non-enzymatic electrochemical sensing (a. doi.org/10.1016/j.trac.2021.116418; b. doi.org/10.1016/j.microc.2020.105599; c. doi.org/10.1007/s00604-022-05612-y).  

Reply: One of the suggested references was cited as reference number 30

Reviewer 3 Report

Comments and Suggestions for Authors

see attached file

Comments on the Quality of English Language

Minor editing of English language required

Author Response

Ms. Sara Strunjas
Assistant Editor

Sensors

The authors are grateful for the revision and recommendations to improve the quality of the manuscript titled “Electrochemical determination of dopamine with a carbon paste-lanthanum (III) oxide micro-composite electrode. Effect of surfactant on selectivity”

The changes in the manuscript were highlighted with yellow color.

Review 3

The manuscript sensors-3144464 “ Electrochemical determination of dopamine with a carbon paste-lanthanum (III) oxide micro-composite electrode. Effect of surfactant on selectivity” presents work that may be of interest because in my opinion, using a commercial product for the application development of a new sensor can be advantageous from the point of view of cost and sensor reproducibility. However, there are several points that need to be clarified

1     The introduction lacks, in my opinion, an overview of the presence in the literature of electrochemical sensors capable of detecting dopamine and its precursors. This makes it clear how important the study you are presenting is (e.g. 10.1016/j.mtcomm.2023.106036, 10.1016/j.jelechem.2024.118129, etc.). Also at the end you provide a list of sensors found in the literature with lanthanum but lacking in my opinion is a comment comparing these with your work

Reply: The authors attempted to present in the introduction the latest reports on PD detection with different types of electrodes and at the end in table 3 different modified electrodes are mentioned besides the lanthanum oxide electrode. The use of surfactants has been added in the introduction since their use in electrochemistry had been mentioned very little, as well as in section 3.5. The discussion in table three was improved.

  1. Even if lanthanum oxide you use is commercial I would include some morphological characterisation. You already include an eds (figure 3) so why not include a sem or even an FTIR of the powder or even the modified electrode?

Reply: A SEM image has been added to the text to provide additional morphological characterization.

3     Figure 1 shows the response of your sensor to DP. Why process at pH 3 since subsequent measurements are all conducted at pH 7?

Reply: At the beginning, an exploratory study is carried out, then it is verified with other measurements to see if that initial value is the best.

4     Given the scan rate measurements in the presence of the redox system [Fe(CN)6]3+/4+ you can indicate whether your sensor works according to a diffusion or adsorption process.

Reply: As for the hexacyanoferrate system, it can be said that it is by diffusion since the Randles-Sevcik equation uses the square root of the scan rate, but it is possible that for DP it is different and that is why it was also studied in section 3.3.

5             I would insert the Randles-Sevcik equation.

Reply: Randles-Sevcik equation was inserted. The authors did not insert it prior because it is a well-known equation and in previous works reviewers have requested to remove it.

6             The pH study should be better commented on. Report your results and comment on those in the literature but do not describe what is observed from the analysis. You have to say which pH value for you is the best to use and why. I would also add a graph showing how current intensity varies with respect to pH.

Reply: The explanation of the pH study has been improved. We chose not to add a graph of current intensity vs. pH to avoid cluttering the text, but the pH analysis has been clarified, and the graph was updated to better highlight the pH values.

7             You must include error bars in all graphs.

Reply: Error bars have been added to the calibration curves in Figures 7 and 9. where they were measured in duplicate with the same electrode. The others were measured using different electrodes but were not duplicated for error bars as they were not necessary for evaluating repeatability and stability at that stage.

8             In my opinion, Figure 6, although discussing important electrochemical parameters, should be moved to supporting information. I would only keep figure 6f to which would add a histogram perhaps with error bars and RSD (or SD) calculations.

Reply: It is a good suggestion. However, the authors consider that presenting the information in text makes it easier for readers to understand and interpret. Standard deviation and coefficient of variation calculations were added to the text

9             Why did you not make EIS measurements to study the electrode?

Reply: The authors attempted the EIS measurements; however, they were inconclusive due to issues with the EIS module. As a public university, we only count with limited equipment resources, with only one potentiostat equipped for EIS.

10          The optimal parameters used for SWV should in my opinion be included in the section ‘2.2 Measurements’ and not at the beginning of section 3.4

Reply: Optimal parameters for SWV have been included in section 2.2. They were mentioned in section 3.4 it was where it was studied.

11          Section ‘3.4 Calibration curve, detection limits and reproducibility’. I do not see any data described showing the reproducibility of your sensor.

Reply: This is a great observation. Additional details on reproducibility and sensor performance have been included in section 3.4.

12          similar to 11

13          Page 10 there is a bold part to remove.

Reply: The authors were unable to identify the bold part

14          I don't understand why you would do a study on a sensor using a pH 7 and then use a pH 2 with an interferer. If you want to eliminate the effect of the interferer, then all the work should have been studied at pH 2. You cannot use an optimum pH to determine LOD and get better responses but then in the presence of an interferer (as is normal in a real sample application) I have to change pH and then all the previous data is no longer useful.

Reply: Thanks for pointing out this inconsistency. We acknowledge the mistake regarding the pH values in Figures 8D and 8E. Corrections have been made to reflect the appropriate pH conditions (pH 7.0).

15          Figure 9 should be better organised. In my opinion there is too much confusion. You could separate the figures for example.

Reply: Figure 9 has been reorganized to reduce confusion and highlight the relevant data more clearly

16          In the caption of Figure 9, you write ‘Figure 9: Square wave voltammograms for urine sample 1 (Table 2) using the LaOx/CPE, with the calibration plot inserted.’ Where is this Table 2 with the urine samples? Table 2 in your manuscript shows something else.

Reply: The mention of Table 2 in the Figure 9 caption was an oversight as the table was removed from the manuscript. The caption has been corrected accordingly.

17          It lacks data on repeatability, response stability over time and sensor reproducibility

Reply: Data on repeatability, response stability, and sensor reproducibility have been included in sections 3.4 and 3.5.

Round 2

Reviewer 1 Report

Comments and Suggestions for Authors

The authors have well addressed my concerns and the manuscript in its current form can be accepted without further revision.

Comments on the Quality of English Language

no

Reviewer 3 Report

Comments and Suggestions for Authors

The authors answered all questions sufficiently.